# Can the delayed effects of climatic oscillations have a greater influence on global fisheries compared to their immediate effects?

**Sandipan Mondal[1,2], Aratrika Ray[1], Malagat Boas[1,3], Sawai Navus[1], Ming-An Lee[1,2,4]\*, Subhadip Dey[5,6], Koushik Kanti Barman[7]**

**1** Department of Environmental Biology and Fishery Science, National Taiwan Ocean University, Keelung City, Taiwan, **2** Center of Excellence for the Oceans, National Taiwan Ocean University, Keelung City, Taiwan, **3** Department of Fisheries and Marine Resources, Papua New Guinea University of Natural Resources and Environment, Kokopo, Papua New Guinea, **4** Doctoral Degree Program in Ocean Resource and Environmental Changes, National Taiwan Ocean University, Keelung, Taiwan, **5** Agricultural and Food Engineering Department, Indian Institute of Technology Kharagpur, Kharagpur, India, **6** Deutsches Zentrum für Luft- und Raumfahrt, Wessling, Germany, **7** Center of Excellence for the Ocean Engineering, National Taiwan Ocean University, Keelung City, Taiwan

\* malee@mail.ntou.edu.tw

**Data Availability Statement:** For access to the data used in this study, please access to the following doi: DOI: 10.5061/dryad.jsxksn0jt.

## Abstract

Climatic oscillations affect fish population dynamics, ecological processes, and fishing operations in maritime habitats. This study examined how climatic oscillations affect catch rates for striped, blue, and silver marlins in the Atlantic Ocean. These oscillations are regarded as the primary factor influencing the abundance and accessibility of specific resources utilized by fishers. Logbook data were obtained from Taiwanese large-scale fishing vessels for climatic oscillations during the period 2005–2016. The results indicated that the effect of the Subtropical Indian Ocean Dipole on marlin catch rates did not have a lag, whereas those of the North Atlantic Oscillation, Atlantic Multidecadal Oscillation, Pacific Decadal Oscillation, and Indian Ocean Dipole had various lags. Pearson's correlation analysis was conducted to examine the correlations between atmospheric oscillation indices and marlin catch rates, and wavelet analysis was employed to describe the influences of the most relevant lags. The results indicated that annual atmospheric fluctuations and their lags affected the abundance and catchability of striped, blue, and silver marlins in the study region. This, in turn, may affect the presence of these species in the market and lead to fluctuations in their prices in accordance with supply and demand. Overall, understanding the effects of climatic oscillations on fish species are essential for policymakers and coastal communities seeking to manage marine resources, predict changes in marine ecosystems, and establish appropriate methods for controlling the effects of climate variability.

## Introduction

Climatic oscillations, often referred to as climatic cycles or fluctuations, have a major effect on ocean conditions and dynamics [1]. These oscillations are inherent patterns that repeat across

**Funding:** National Science & Technology Council of Taiwan (NSTC), NSTC 112-2811-M-019-004.

**Competing interests:** The authors have declared that no competing interests exist.

time periods ranging from years to millennia while influencing various oceanic factors, such as the sea surface temperature (SST), ocean currents, precipitation, and sea level [2]. During the El Niño southern oscillation (ENSO), the SST in the tropical Pacific increases considerably, resulting in widespread warming, whereas during the La Niña a southern oscillation, the opposite effect occurs [3]. The Pacific Decadal Oscillation (PDO) influences the intensity and trajectory of major ocean currents, such as the Gulf Stream [4]. The North Atlantic Oscillation (NAO) is the primary cause of fluctuations in the North Atlantic (NA) region, occurring over time periods ranging from a single season to often 20–30 years [5]. The NAO is associated with fluctuations in several climatic factors, such as wind velocity and direction, as well as changes in air temperature and precipitation. Another common indicator of decadal climatic variability in the Atlantic is the Atlantic Multidecadal Oscillation (AMO), which is associated with changes in the Atlantic Meridional Overturning Circulation through the NAO [6]. According to recent studies, the distribution and abundance of Atlantic bluefin tuna (BFT, *Thunnus thynnus Linnaeus*, 1758) are linked to changes in the environment caused by both the AMO and NAO. During positive AMO phases, when the ocean is warm, BFT are more abundant (the eastern Atlantic population), whereas during negative AMO phases, when the ocean is cold, BFT are less abundant (western Atlantic population) [7]. The effects of these climatic oscillations on ocean conditions are complex and may vary depending on the phase and intensity of the oscillation as well as on interactions with other climate variables. Climatic oscillations may have a delayed effect on ocean conditions because their abnormalities continue for many months or even years after the oscillation occurred [8]. Although multiple studies have indicated a correlation between past climatic oscillations and current ocean conditions, no studies have yet examined how climatic oscillations specifically affect migratory pelagic species in the Atlantic Ocean (AO) or have identified the prospective adjustments that may be possible due to the current estimates of increasing global temperature [9].

In the AO, climatic oscillations affect local weather patterns, oceanographic conditions, and ecosystems [8]. In the 1900s, Gilbert Thomas Walker scientifically quantified fluctuations in climate conditions in the NA region and established the term NAO to describe this phenomenon. The NAO index reflects atmospheric pressure fluctuations between the Icelandic Low and the High over the Azores archipelago. The NAO index affects ocean conditions such as heat content, SST, gyre circulation, mixed layer depth (MLD), salinity, deep water formation, and sea ice cover, making it useful for analyses of marine ecosystem variability [10]. The AMO index is defined as the internal mode of climatic variability for a time series of the averaged SST in the NA. Multiple empirical and modeling studies have indicated that the AMO influences summer precipitation in North America and Europe, Atlantic hurricane activity, and other climatic features [11]. According to Alheit et al. (2014) [12], the AMO also affects the behavior of small pelagic fish species and results in changes in ecological regimes in the eastern NA and Central Atlantic (CA). The NAO and AMO work in a complementary fashion. During the period between the mid-1960s and mid-1990s, which represented a warm phase of the AMO, the winter NAO index shifted from extremely negative to extremely positive values [10].

A large body of literature suggests that marine biological processes can be best explained by climatic oscillations, rather than individual climate variables, because these oscillations are weather packages that influence the responses of ecosystems [13]; they affect many weather variables simultaneously and sometimes distant regions. Interactions between the three tropical ocean basins substantially influence the formation of both short-term and long-term climate variability [14]. According to Heffernan et al. (2014) [15], teleconnections are the relationships between the characteristics of an ecosystem and distant climate fluctuations. The NAO index is a unique macroscale variable that combines many climatic variables and

averages meteorological conditions over time and location. This index may therefore play a role in regional ocean–atmosphere interactions with complex regional consequences, similar to the more complex connection between the ENSO and Southern Oscillation Index in the Pacific Ocean (PO), which affects the entire world [10]. A key teleconnection is that between climatic oscillations and fishery yields because many fishery resources are synchronized with the NAO [16] and other types of large-scale climate variability [7, 12]. More specifically, climatic oscillations affect the recruitment, catchability, and body condition of target species [12, 16]. In addition, the ENSO, NA climate, and East Asia are affected by climate variability in the Indian Ocean (IO) [17] through atmospheric teleconnections and oceanic processes [18]. In a recent study, Yang et al. (2022) [19] examined how the PO influences the NA–IO warming chain, thereby providing insights into the worldwide systemic connection between the three regions through evolution of the Walker Circulation. The Indian Ocean Dipole (IOD), a major mode of interannual climatic variability in the tropical IO, is characterized by an east-to-west dipole-like SST anomaly pattern. In the tropical AO, the positive phase of the IOD triggers westerly wind anomalies, thereby weakening easterly trade winds and resulting in warm SST anomalies [20]. A tropical IOD is triggered by the Subtropical Indian Ocean Dipole (SIOD), a second source of SST fluctuations in the subtropical IO [21] and created by wind, evaporation, and SST anomalies linked to wind circulation anomalies [22]. Because the SIOD affects the climate of rim countries, understanding its triggers is essential [23].

Marlins belong to the family Istiophoridae and play a major ecological role in marine ecosystems [24]. As apex predators, they exert control on the amount and behavior of their prey and hence influence the structure and dynamics of the food web. In addition to having ecological importance, marlins are often used for their meat and fins, which reflects economic importance [25]. Genetic divergence analysis indicated the merging of two separate species of marlins into a single species in the Indo-Pacific and AO, with the analysis findings supported by the results of tagging experiments [26]. This study examined the relationship between climatic oscillations, including their lagged variants, in the AO and the rates of marlin catches (Striped marlin—*Kajikia audax*, 1887; Blue marlin—*Makaira nigricans*, 1802; Silver marlin—*Kajikia albida*, 1860) by Taiwanese longline fishing vessels. The striped marlin, a renowned predator in recreational fishing, is known for its elongated body, vivid blue and silver hues, and large dorsal fin. Its vertical stripes become more prominent when stimulated or engaged in hunting. The blue marlin, with its vibrant cobalt blue upper body and shiny white underbelly, is a large predator with a long, pointed beak and a tall, curved dorsal fin. Its main food is fish and cephalopods. The silver marlin, with its streamlined, elongated body, cobalt blue dorsal surface, and pristine silvery ventral region, is a sought-after species for its dexterity and swiftness. The Atlantic white marlin's dorsal fin is conspicuous and curved, contributing to its sleek and streamlined appearance. The species primarily feeds on tiny fish and squid, making it an effective predator in its marine environment. These species are highly valued in recreational fishing for their dexterity and robustness. The specific rationale for choosing only these three marlin species was based on the fishery data only provided by the Overseas Fisheries Development Council of Taiwan, which exclusively consisted of data pertaining to these three marlin species. In addition, these three marlin species are vital species in the Atlantic Ocean, playing a crucial role as apex predators, regulating prey populations, supporting commercial and recreational fishing industries, and generating revenue through tourism. They also symbolize maritime heritage and local traditions. Because of the high demand for and commercial value of marlins, marlin fisheries substantially contribute to the seafood industry worldwide and provide ample job opportunities and income for millions of people. Despite these benefits, no studies have yet comprehensively examined the effects of climatic oscillations and ocean conditions on the catch rates and distribution of marlins in the AO [8]. In addition, there are

concerns about the stock condition of marlins, in the Atlantic Ocean, as there are indications of overfishing and a decrease in population numbers [27, 28]. As per the International Commission for the Conservation of Atlantic Tunas (ICCAT), the populations of both blue and striped marlins have been excessively exploited and are currently below levels that can be sustained in the long term [29, 30]. Blue marlin populations are significantly reduced due to heavy fishing pressure from commercial longline fisheries, primarily targeting tuna and swordfish, often caught incidentally as bycatch. Hence, in order to guarantee the long-term conservation of fishery resources, additional investigation is necessary to ascertain the impact of climatic oscillations on marlin fisheries due to the ecological and economic significance of marlins. Further research is also required to determine whether oscillations lead to variations in the population and catchability of commercially valuable marlin species, where these variations may influence fish market prices through changes in the supply of fishing products. It has been postulated that climatic oscillations, along with their delayed phases, exert a greater influence on catchability compared to immediate effects. The primary two objectives of this study are: (i) to establish the correlation between the key inter-annual climatic oscillations in the AO, namely the NAO, AMO along with two other large-scale oscillations from Pacific Ocean, namely, PDO and SIOD, having teleconnection effects on AO and the landings of three species of marlins in AO (ii) to utilize climatic oscillations as a proxy for ecological predictions, to establish the conceptual foundations of the relationships between lags and marlin fish catch, and (iii) to understand the level of resilience of fisheries to variability in climate oscillations.

## Methods

### Data collection

**Marlin fishery data.**   Logbooks of Taiwanese large-scale longline fishing vessels, which are deep-water fishing vessels with volumes exceeding 100 gross registered tons and lengths exceeding 24 m, were used to collect monthly fishery data for striped, blue, and silver marlins. These logbooks, which covered the period from 2005 to 2016, were obtained from the Overseas Fisheries Development Council. The data covered the area from 1° S to 40° S and from 10° E to 40° W, with resolution of 1° × 1°. The logbooks also included data regarding the year, month, latitude, longitude, number of captures, total catch weight (kg), and number of hooks used. However, they did not include data regarding hook depth or operation duration.

**Climatic oscillation data.**   Data regarding five climatic oscillations, namely the NAO, AMO, PDO, IOD, and SIOD, were obtained for the period 2005–2016. Each climatic oscillation was evaluated in consideration of delays of up to 8 years (yearly temporal resolution). S1 Table details the sources of the climatic oscillation data.

### Data examination

**Interannual fluctuations in catch rates.**   The catch rate for each marlin species was calculated as follows:

$$\text{Catch rate} = \frac{\text{Catch in weight (Catch)}}{\text{Number of hooks deployed (Effort)}} \tag{1}$$

Autoregressive integrated moving average (ARIMA) time series analysis was conducted to evaluate the annual fluctuations in catch rates. ARIMA is a statistical model that incorporates autoregressive and moving average components, as well as differencing, to accurately represent the underlying patterns in time series [31]. Analyses were conducted using R software (version

4.2.3; R Foundation for Statistical Computing, Vienna, Austria) with the "ts" function from the "tseries" package [32] and the "cpt.meanvar" function from the "changepoint" package [33].

**Correlations between catch rate and climatic oscillations.** The Pearson's correlations between the catch rate of each species and the climatic oscillations were analyzed while accounting for the lags or latencies in the oscillations [34]; these analyses were performed in R software (version 4.2.3) with the "cor.test" function from the "corrr" package. A climatic oscillation was analyzed further only if its coefficient of correlation with the catch rate of any marlin species was 0.1 or higher [35]. Pearson's correlation coefficients were calculated as follows:

$$r = \frac{\sum_{i=1}^{n} (X_i - X)(Y_i - Y)}{\sqrt{\sum_{i=1}^{n} (X_i - X)^2} \sqrt{\sum_{i=1}^{n} (Y_i - Y)^2}} \tag{2}$$

where $n$ is the sample size, $i$ represents individual sample points ($X_i$ and $Y_i$), and $X$ and $Y$ are sample mean values.

A higher correlation coefficient indicates collinearity between paired variables. In addition, a Pearson's correlation coefficient of approximately 0.8 indicates collinearity [36]. The three marlin species were discovered to be affected by climatic oscillations for which the correlation coefficient was greater than 0.1, suggesting a nonlinear effect. Therefore, only the time delays of climatic oscillations with a correlation coefficient greater than 0.1 were selected for analysis; the correlation coefficients could range between −1 and +1. According to Verma (2021) [37], the degree of correlation between two variables indicates the extent to which changes in one variable are influenced by changes in the other. This test has the capacity to uncover major correlations between climatic factors and fish capture in various time intervals, indicating that the distribution and quantity of fish, including catches of these fish, depend on ecosystem parameters [38]. Analysis of the correlation structure of a time series may reveal fundamental patterns such as seasonality and trends [39].

**Effect of climatic oscillations on marlin catch rates.** Generalized additive models (GAMs) were employed to determine the effect of climatic oscillations on the catch rates for striped, blue, and silver marlins. GAMs enable the analysis of nonlinear correlations and the identification of complex data patterns by utilizing smooth functions to accommodate any nonlinearity in predictor variables. The results of GAMs are linked to explained deviance, which is a measure quantifying the amount of variability in a dependent variable that a model accounts for. This measure reflects the reduction observed in deviance as compared with a null model. Higher values of explained deviance indicate a greater ability of GAMs to explain the observed variability in the dependent variable [40]. In this study, a GAM was established for each species, with climatic oscillations used as predictor variables and the catch rate used as the response variable [41]. Analyses were conducted using R software (version 4.2.3) with the "smoothing" function from the "mgcv" package. Subsequently, the climatic oscillation factors were ranked on the basis of the amount of explained deviance, generalized cross validation (GCV), and the Akaike information criterion (AIC). GCV is commonly used in offline scenarios to predict unknown parameters in regularized estimators. This technique has gained popularity in machine learning for enhancing the generalization capabilities of regularized kernel-based methods, such as regularization networks, which include spline regression as a specific instance. The AIC is a quantitative measure used to evaluate the outcomes of various models while accounting for both the goodness of fit and the complexity of each model. Lower GCV and AIC values indicate a better fit and higher parsimonious model complexity, respectively. In this study, only the models that exhibited the highest explained deviance and the lowest

AIC value were selected for further analysis. All GAMs were generated using the following mathematical formula:

$$\text{GAM} = (\text{catch rate} + c) \sim s(\text{predictor variable}) \tag{3}$$

where $c$ is a constant (0.1) and $s$ is a smoothing function. For each species, only the climatic oscillation with the most significant effect on the catch was selected as a predictor, as determined by the lowest AIC and highest explained deviance.

**Phase-wise interrelation between catch rate variability and climatic oscillations.** The utilization of wavelet analysis was employed to examine the influence of climatic oscillations on the catch rates of marlins. Wavelet analysis is a commonly used method for examining multiple time series data [42, 43]. Time series decomposition is a technique that allows for the examination of periodic components and their changes over time by representing the data in a space that combines both time and frequency [44]. The method calculates a wavelet coefficient for every position in the dataset by quantifying the correlation between the observed signal (in this instance, the chosen climatic index) and the wavelet. The result is a matrix that includes wavelet coefficients for all conceivable combinations of scale (representing the frequency of the wavelet) and translation (showing the shift of the wavelet along the signal). The Fourier spectrum analysis is the primary method employed to investigate periodic patterns in time series data [45]. However, this method assumes that the time series must be steady, which was not true for the longline fishery data and temperature time series. Wavelet analysis was employed because of its capacity to evaluate time series data without making assumptions about stationarity. Wavelets have the benefit of concurrently analyzing several scales, a capability that Fourier analysis lacks as it only identifies dominant frequencies in time series [44, 46]. The wavelet transform decomposes a time series into its individual time, frequency, and power elements, which are shown in a three-dimensional space using the wavelet power spectrum (WPS) plot. The x-axis in a WPS plot depicts the time series, while the y-axis represents the contribution of frequencies, which is denoted by the term "period" [47]. Power is the quantitative measure of the level of variability in a time series at a particular wavelet. The WPS evaluates the influential and contributory parts of the signal, as well as the less significant ones. Wavelet transformations have become a prevalent method for studying variations, periodicities, and trends in time series over the past decade [48]. Wavelet transforms offer valuable decompositions of the original time series, enhancing the forecasting model's effectiveness by capturing significant information across many resolution levels. In addition, cross-wavelets and phase analysis extend these capabilities to the examination of relationships between two signals [49]. Wavelet coherence can be useful in determining relationships when temporal lags complicate correlation. Two variables are considered coherent when they display synchronized oscillations and maintain constant phase differences throughout time, regardless of whether they are out of phase or in anti-phase. The consistency of phase differences is crucial since it is more likely to indicate a causal relationship that is not attributable to random chance. Moreover, determining the specific phase correlations can provide valuable insights into the possible causes. Similar to other wavelet methods, wavelet coherence also handles complex temporal autocorrelation structures, which are a prevalent characteristic of fisheries time series that can complicate correlation tests [50]. In the present study, the interrelation between the annual fluctuations observed in the catch rate of a marlin species and the annual variability in the climatic oscillation selected for that species was analyzed using cross-wavelet time series analysis [51, 52] in R software (version 4.2.3) with the "wtc" function from the "biwavelet" package [8]. In accordance with Grinsted et al. (2004) [53], the cross-wavelet coherence of two time series, namely the annual catch rate and climatic oscillation variability, was defined as

follows:

$$R_n^2(s) = \frac{\left| S\left[ s^{-1} W_n^{XY}(s) \right] \right|^2}{S\left( s^{-1} W_n^X(s) \right)^2 \cdot S\left[ s^{-1} W_n^Y(S) \right]^2} \tag{4}$$

where $W$ is the wavelet transform of the time series, $S$ is a smoothing operator used to calculate the average values, and $X$ and $Y$ are two distinct time series, namely the catch rate and climatic oscillation, respectively.

**Cumulative effect of climatic oscillations on catch rates.** GAMs were used to evaluate the cumulative effect of climatic oscillations, with different time delays, on the catch rate of each species. Separate models were constructed for each species by using all the potential combinations of climatic oscillation factors identified in the GAM analysis of this study. Before modeling was performed, pairs were formed, and the collinearity of the effects of climatic oscillations on catch rates was evaluated by calculating the variance inflation factor (VIF) in R software (version 4.2.3) with the "vif" function from the "car" package. Only climatic oscillations with a VIF value of 5 or less [54] were used for model construction. GAMs were individually constructed for each species by using pairs of climatic oscillation variables in R software (version 4.2.3) with the "smoothing" function from the "mgcv" package. Before the models were constructed, VIF analysis was conducted. All paired GAMs were generated as follows:

$$\text{GAM} = (\text{catch rate} + c) \sim \text{s}(\text{climatic oscillation 1}) + \text{s}(\text{climatic oscillation 2}) \tag{5}$$

## Results

### Annual catch rate variability

From 2005 to 2016, major fluctuations were observed in the catch rates for striped, blue, and silver marlins. The catch rates for striped and blue marlins exceeded those for silver marlins throughout the study period. The year of the highest catch rate was different for each species. In the case of striped marlins, the highest catch rate (approximately 30 kg/$10^4$ biomass) was recorded in the winter of 2014–2015, followed by 2005 and 2007 (approximately 20 kg/$10^4$). In the case of blue marlins, the highest catch rates were recorded in 2007 (64 kg/$10^4$ biomass) and 2009 (78 kg/$10^4$ biomass). In the case of silver marlins, the highest catch rate was recorded in 2010 (>25 kg/$10^4$ biomass). After 2010, specifically between 2011 and 2016, the overall catch rates for silver marlins were much lower than before 2010, with almost no fish caught in 2013, 2014, or 2016. Similar fluctuations were discovered in the catch rates for striped and blue marlins from 2015 to 2016, with high catch rates particularly in 2005, 2007, 2009, 2009, 2012, and 2014. As illustrated in Fig 1, the catch rates for blue marlins were higher than those for the other species throughout the study period.

### Correlation between catch rates and climatic oscillations

Table 1 presents the correlations between the striped, blue, and silver marlin catch rates and climate variables, when considering various time lags. Strong correlations were discovered between the striped marlin catch rate and 3- and 6-year-lagged NAO; 0-, 1-, 3-, 5-, and 6-year-lagged AMO; 2–4- and 6–8-year-lagged PDO; 1-, 2-, 4-, and 5-year lagged IOD; and 0-, 1-, and 3–5-year-lagged SIOD. The strongest negative correlation ($r = -0.651$) was that with the 1-year-lagged IOD, whereas the strongest positive correlation ($r = 0.678$) was that with the nonlagged SIOD. The NAO and AMO exhibited the strongest correlations with the catch rate for a time lag of 6 years, whereas the IOD and PDO exhibited the strongest correlations with

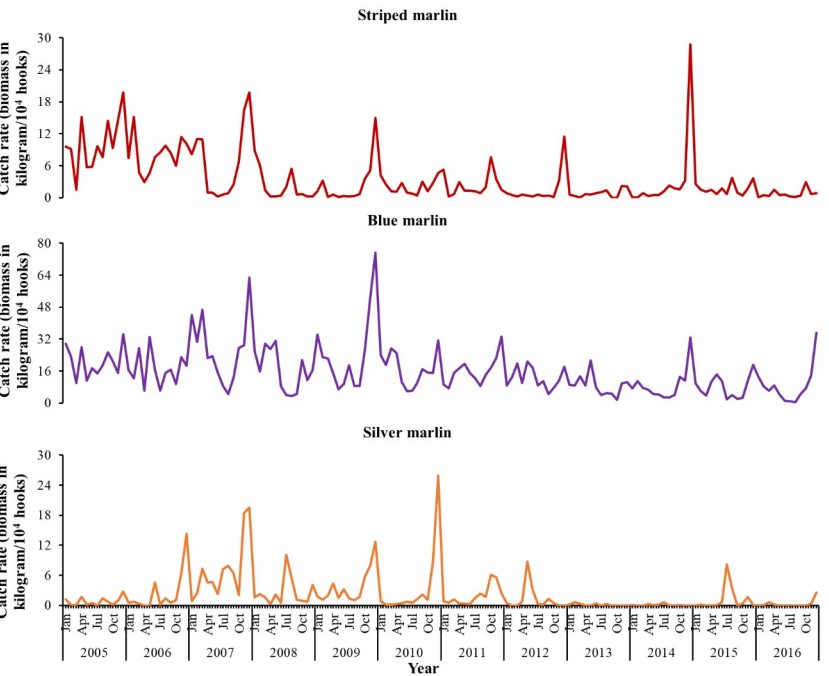

**Fig 1. Annual fluctuations in the catch rates for striped, blue, and silver marlins during 2005–2016.**

the catch rate for time lags of 1 and 3 years, respectively. By contrast, the SIOD exhibited the strongest correlation with the catch rate for a time lag of 0 years (immediate effect).

The blue marlin catch rate exhibited a significant correlation with the NAO at all time lags except 2–3 years. It was also significantly correlated with the AMO at all time lags except 3–4

**Table 1. The correlation between the catch rates of three marlin species and various climate oscillations and their time delays.**

| Species | Striped Marlin | | | | | Blue Marlin | | | | | Silver Marlin | | | | |
|---|---|---|---|---|---|---|---|---|---|---|---|---|---|---|---|
| Lag years | *NAO* | *AMO* | *PDO* | *IOD* | *SIOD* | *NAO* | *AMO* | *PDO* | *IOD* | *SIOD* | *NAO* | *AMO* | *PDO* | *IOD* | *SIOD* |
| 0 | -0.063 | **0.455** | 0.124 | -0.178 | **0.678** | **-0.306** | **0.576**\* | **-0.428** | 0.187 | **0.646**\* | -0.194 | **0.433** | -0.328 | **0.462** | **0.608**\* |
| 1 | 0.014 | **0.308** | 0.247 | **-0.651**\* | **0.514**~ | **-0.316** | 0.483 | -0.196 | -0.024 | **0.837**\*\*\* | -0.274 | **0.383** | -0.065 | 0.159 | **0.704**\* |
| 2 | 0.224 | -0.011 | **0.526**~ | -0.452 | 0.109 | -0.005 | 0.353 | 0.351 | -0.252 | **0.526**~ | -0.141 | **0.369** | 0.147 | **-0.467** | **0.401** |
| 3 | **0.404**~ | -0.463 | **0.599**\* | -0.223 | **-0.345** | 0.277 | -0.071 | **0.688**\*\* | -0.253 | **0.377** | 0.305 | 0.041 | **0.518**~ | -0.231 | **0.569**~ |
| 4 | 0.184 | -0.291 | **0.337** | -0.425 | -0.386 | **0.348** | -0.042 | **0.789**\*\* | **-0.671**\* | 0.052 | **0.424** | -0.087 | **0.807**\*\*\* | -0.305 | 0.102 |
| 5 | 0.211 | **-0.426** | -0.021 | **-0.329** | **-0.398** | **0.468**~ | **-0.372** | **0.591**\* | **-0.568**\* | -0.304 | 0.179 | -0.152 | **0.621**\*\* | **-0.568**~ | **-0.524**~ |
| 6 | **0.492**~ | **-0.535**~ | **-0.532**~ | -0.202 | 0.011 | **0.498**~ | **-0.356** | 0.274 | **-0.497** | -0.151 | 0.219 | -0.227 | **0.316** | -0.459 | -0.219 |
| 7 | 0.144 | -0.235 | -0.288 | -0.241 | -0.187 | **0.302** | **-0.553**~ | 0.122 | **-0.451** | **-0.623**\* | **0.443** | **-0.631**\* | -0.031 | -0.192 | **-0.389** |
| 8 | -0.158 | 0.208 | **0.369** | **0.471** | -0.268 | 0.283 | **-0.402** | -0.069 | 0.027 | **-0.333** | **0.558**\* | **-0.664**\* | -0.492 | -0.227 | -0.152 |

**Note:** Correlation values > 0.3 eliminated before the next step and correlations that were chosen are shown in bold. Significance levels:

\*\*\* 0.001,

\*\* 0.01,

\* 0.05,

~ 0.1.

NAO–North Atlantic oscillation; AMO–Atlantic multidecadal oscillation; PDO–Pacific decadal oscillation, IOD–Indian Ocean dipole; SIOD–Sub-tropical Indian ocean dipole

years, with the PDO at time lags of 0 and 2–5 years, with the IOD at time lags of 4–7 years, and with the SIOD at all time lags except 4 years. The strongest positive and negative correlations were those with the 1-year-lagged SIOD ($r = 0.837$) and 4-year-lagged IOD ($r = −0.671$), respectively. The strongest correlation between the NAO and the blue marlin catch rate was that after 6 years, whereas the strongest correlations between the PDO and IOD and the blue marlin catch rates was that after 4 years. The strongest correlations between the AMO and SIOD and the blue marlin catch rates was that after 0 years.

The silver marlin catch rate was significantly correlated with the 3-, 4-, 7-, and 8-year-lagged NAO. Additionally, this rate exhibited strong correlations with the 0–2-, 7-, and 8-year lagged AMO; 0-, 2-, and 4–6-year-lagged IOD; and 0–3- and 5–7-year-lagged SIOD. The strongest positive and negative correlations were those between the silver marlin catch rate and the 4-year-lagged PDO ($r = 0.807$) and 8-year-lagged AMO ($r = −0.664$), respectively. The strongest correlations of the silver marlin catch rate with the NAO and AMO were those after 8 years, followed by the PDO after 4 years. The strongest correlation between the silver marlin catch rate and IOD was that after 5 years, whereas the strongest correlation between the silver marlin catch rate and the SIOD was that after 0 years.

## Effect of specific climatic oscillations on catch rates

Table 2 (detailed in S2 Table) present the climatic oscillations that resulted in the greatest variations in the catch rates for striped, blue, and silver marlins, as identified by GAM analysis. The striped marlin catch rate was significantly affected by the 0-year-lagged SIOD and AMO (immediate effect), 2-year-lagged PDO, 3-year-lagged NAO, and 1-year-lagged IOD. These climatic factors accounted for 99.7%, 86.7%, 91.4%, 86.7%, 76.3%, and 42.5%, respectively, of the deviation observed in the striped marlin catch rate. The striped marlin catch rate was also significantly affected by the 3-year-lagged AMO (75.2%). The oscillations that had the least or no significant effect for the striped marlin were the 6-year-lagged NAO, the 1-year-lagged AMO, the 4-year-lagged PDO, the 2-year-lagged IOD, and the 4-year-lagged SIOD (S2 Table).

The blue marlin catch rate was significantly affected by the 5-year-lagged AMO, non-lagged SIOD, 3-year-lagged PDO, 7-year-lagged IOD, and 4-year-lagged NAO. These factors accounted for 96.6%, 94.9%, 74.2%, 66.3%, and 43.8%, respectively, of the deviation in this

**Table 2. The time lags of major climate oscillations that have the most impact on the variation in marlin catch rates, as identified by GAM analysis.**

| Striped marlin | | | | Blue marlin | | | | Silver marlin | | | |
|---|---|---|---|---|---|---|---|---|---|---|---|
| *Selected oscillation lag* | *Deviance explained (%)* | *GCV* | *AIC* | *Selected oscillation lag* | *Deviance explained (%)* | *GCV* | *AIC* | *Selected oscillation lag* | *Deviance explained (%)* | *GCV* | *AIC* |
| NAO3* | 76.3 | 8.26 | 56.43 | NAO4~ | 43.8 | 43.29 | 79.71 | NAO8~ | 73.5 | 2.78 | 45.32 |
| AMO* | 86.7 | 5.25 | 50.21 | AMO5~ | 96.6 | 23.05 | 57.01 | AMO8* | 88.7 | 1.95 | 38.17 |
| PDO2* | 91.4 | 4.61 | 46.49 | PDO3* | 74.2 | 21.27 | 70.91 | PDO* | 96.8 | 1.41 | 7.11 |
| IOD1* | 42.5 | 6.83 | 58.74 | IOD7* | 66.3 | 23.11 | 72.61 | IOD5* | 96.8 | 2.14 | 28.36 |
| SIOD* | 99.7 | 0.88 | 12.98 | SIOD** | 94.9 | 10.11 | 57.06 | SIOD* | 99.6 | 0.32 | 2.98 |

Significance levels:

*** 0.001,

** 0.01,

* 0.05,

~ 0.1.

NAO–North Atlantic oscillation; AMO–Atlantic multidecadal oscillation; PDO–Pacific decadal oscillation, IOD–Indian Ocean dipole; SIOD–Sub-tropical Indian ocean dipole

catch rate. The blue marlin catch rate was also significantly affected by the 7- and 8-year-lagged AMO (64.7% and 58%, respectively); 4-, and 0-, year-lagged PDO (62.3%, and 55.6%, respectively); 4- and 6-year-lagged IOD (45%, and 42.3%, respectively); and 1- and 7-year-lagged SIOD (70.1%, and 38.9%, respectively). Of these predictors, the 1-year-lagged SIOD was the most significant ($p = 0.001$), with the most favorable GCV or AIC value. The oscillations that had the least or no significant effect for the blue marlin were the 7-year-lagged NAO, the 6-year-lagged AMO, the 5-year-lagged PDO, the 5-year-lagged IOD, and the 8-year-lagged SIOD (S2 Table).

In the case of silver marlins, the highest deviance was explained (99.6%) by the 0-year-lagged SIOD. In addition, the 0-year-lagged SIOD exhibited high deviance for time lags of 1, 5, and 3 years (80.2%, 59.8% and 32.5%, respectively). The second-highest deviance (96.8%) was explained by the non-lagged PDO and 5-year-lagged IOD. In addition, the 2-year-lagged IOD was found to have a significant effect ($p = 0.05$). Other time lags of PDO were associated with high values of deviance after 4, 5, and 3 years (93.1%, 38.6% and 26.9%, respectively) (S2 Table). Moreover, the 8-year-lagged AMO explained 88.7% of the deviance, the 7-year-lagged AMO was associated with high deviance (61.9%), and the 8-year-lagged NAO explained 73.5% of the deviance.

In addition, the 0-year-lagged SIOD notably had significant influences on all three marlin species, unlike the other oscillations (NAO, AMO, IOD, and PDO), which exerted their effects only after several years. S1 Fig presented the partial effect plots of the marlins with the most significant lag of each climatic oscillation.

## Phase-wise interrelation between catch rate variability and climatic oscillations

This study identified 10 phases of significant correlations between the striped marlin catch rate and 3-year-lagged NAO: positive correlations in 2006–2016 and a negative correlation in 2012. In 2012–2013, synchronicity between the two time series was discovered, with one variable presumably influencing the other after a specific time lag. The phases of the relationship between the NAO and catch rate lasted between 4 and 8 years. In 2014–2015, a negative correlation was observed between the 0-year-lagged AMO and a phase of significant interconnectedness, whereas in 2007–2015, several phases of positive correlations were discovered, with the strongest correlation in 2009–2014. In addition, significant correlations with the SIOD, with no time lags, were found for 2007–2015, with both positive and negative phases. The most significant negative phases were in 2007–2009, 2011–2012, and 2014–2015, and two positive phases occurred in 2008 and 2012–2013. Significant correlations between the striped marlin catch rate and 2-year-lagged PDO were found in 12 discrete time intervals. Negative correlations were significantly more frequent than positive correlations. Specifically, positive correlations were discovered in 2011–2012 and 2013–2014. In 2012–2015, a strong correlation between the two time series was found, with one variable presumably influencing the other after a specific time lag. The strongest negative correlations were in 2007–2010, 2009–2011, and 2013–2015. These correlations presumably occurred because of the link between the NAO and catch rate for a duration of 4–8 years. In 2014–2015, a negative correlation was found between the AMO and a phase of significant interconnectedness, with no time lags. Multiple phases of positive correlations in 2007–2015 were also discovered, with the strongest correlation during the period between 2009 and 2014 (Fig 2a).

Significant correlations between the blue marlin catch rate and 4-year-lagged NAO were found in eight phases: two positive correlations in 2006–2007 and 2013–2015 and six negative correlations throughout. The correlations between the NAO and catch rate persisted for 4–8

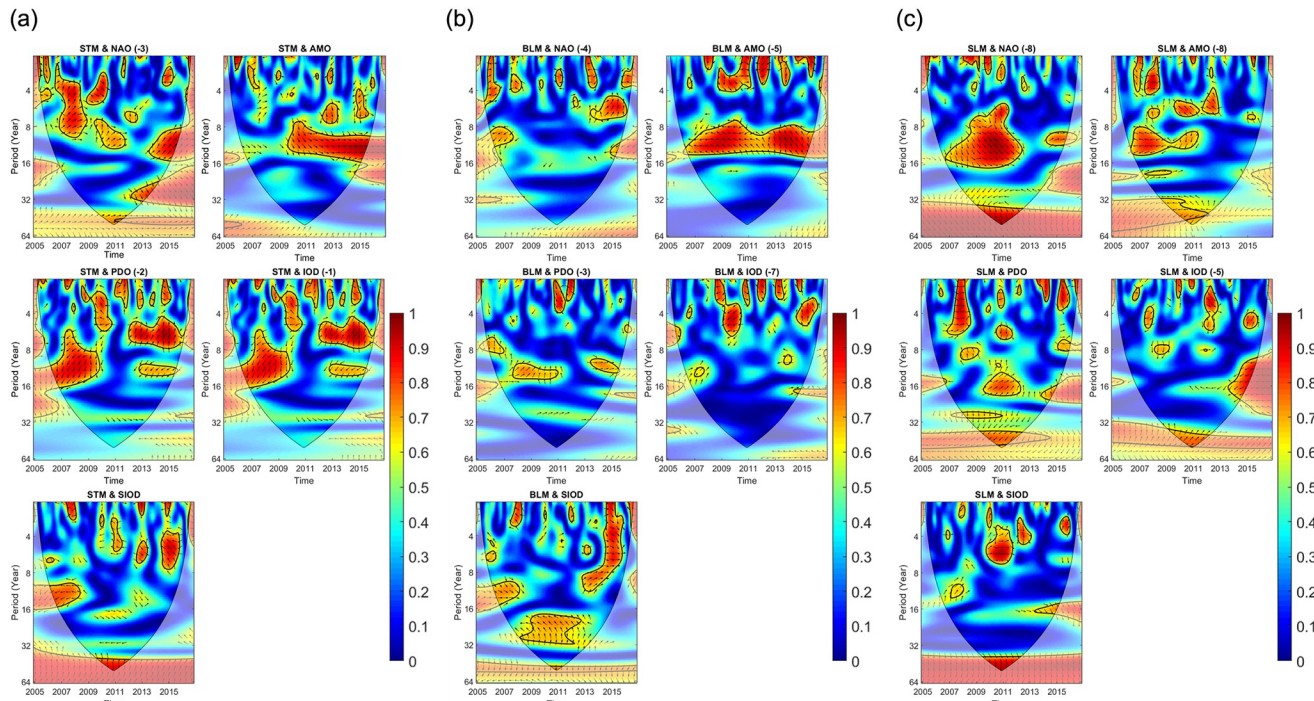

**Fig 2. Relationship between climatic oscillation variability and the catch rates for (a) striped, (b) blue, and (c) silver marlins, as identified through GAM analysis.** The *y*-axis represents the period from 2005 to 2016. The legend ranges from 0 (blue, lowest) to 1 (red, highest), indicating the level of interrelation.

years. Given a time lag of 5 years, the strongest negative correlation was that between the AMO and a phase of significant interconnectedness for almost 8 years from 2007 to 2015, during the antiphase relationship in 2007–2011 and delayed relationship in 2012–2015, with one variable presumably influencing the other after a specific time lag. Multiple phases of positive correlations were discovered in 2010–2011 and 2013–2015, with a minor negative phase observed in 2012. The effect of lagging between time series was clearly observed in 2008 and 2009–2010. In the absence of time lags, significant correlations were found between the SIOD and both positive and negative phases in 2007–2015. The most notable upturns were observed in 2008–2013, with positive in-phase correlations observed in 2008 and 2013–2015. By contrast, antiphase relationships were found in 2008–2009 and 2008–2013. This study identified five distinct time windows when the striped marlin catch rate was significantly correlated with the 3-year-lagged PDO. Positive correlations were also identified between 2013 and 2015. Upturns and downturns were more common than were positive or negative correlations. Upturns occurred between the two time series in 2006–2007 and 2007–2011, whereas a single downturn occurred in 2011–2012, indicating that one variable presumably influenced the other after a specific time lag. Multiple phases of negative correlations between the 7-year-lagged IOD and the blue marlin catch rate were observed in 2006–2007, 2009–2010, and 2012–2013, and a single small positive phase in 2010–2011 was found, with a cycle of less than 4 years and one upturn in 2013–2015 (Fig 2b).

The silver marlin catch rate and 8-year-lagged NAO were found to be significantly correlated in seven unique phases. The strongest positive correlation was that in 2009–2013, following a downturn in 2007–2009. The correlation between the NAO and catch rate persisted for 4–8 years. This study determined three strong negative correlations between the silver marlin

catch rate and 8-year-lagged AMO for nearly 4–8 years in 2006–2012 and 2012–2013. A downturn was also observed in the later phase of negative correlations, specifically in 2009–2011, indicating a cause-and-effect relationship with a time lag. The phases of positive correlations were in 2007 and 2008–2009. In the absence of a time lag, a strong positive correlation was found between the SIOD and silver marlin catch rate in 2010–2011. Small upturn phases in 2007–2008, 2010, and 2013–2015 and two downturns in 2009–2012 were found. The striped marlin catch rate and nonlagged PDO were significantly correlated in 14 specific periods, with positive correlations in 2007–2009 and 2009–2012. Three small upturn phases occurred in 2008–2009 and 2014, with downturn phases in 2008–2012 and 2015. Two phases of negative correlations between the silver marlin catch rate and 7-year-lagged IOD were found for 2008, 2010, 2011, and 2013–2015, with an intermediate positive correlation in 2006, 2009–2012, and 2015 (Fig 2c).

## Cumulative effect of climatic oscillations on marlin catch rates

The examination of the cumulative effect of climatic oscillations on marlin catch rates, when considering various time lags, revealed no collinearity effect in VIF analysis (S2 Fig). For blue marlins, the lowest GCV and AIC values (6.89 and 41.59, respectively) were those for the combination of the 5-year-lagged AMO and 0-year-lagged SIOD (Table 3). For silver marlins, the lowest GCV and AIC values (0.17 and 10.21, respectively) were those for the combination of the 8-year-lagged AMO and 0-year-lagged SIOD. For striped marlins, the lowest GCV and AIC values (2.79 and 26.04, respectively) were those for the combination of the 0-year-lagged AMO and 2-year-lagged PDO.

## Discussion

Climatic oscillations are patterns of atmospheric and sea-surface phenomena that influence average weather conditions and other climatic features over large areas; they typically involve

**Table 3. Evaluating models based on the most significant oscillation variables that contribute to the catch rate of each marlin species.**

| Striped marlin | | | Blue marlin | | | Silver marlin | | |
|---|---|---|---|---|---|---|---|---|
| **Models** | **GCV** | **AIC** | **Models** | **GCV** | **AIC** | **Models** | **GCV** | **AIC** |
| NAO3, AMO** | 4.04 | 27.16 | NAO4, AMO5*** | 34.51 | 77.64 | NAO8, AMO8*** | 1.21 | 31.56 |
| NAO3, PDO2*** | 5.44 | 46.56 | NAO4, PDO3*** | 7.91 | 43.21 | NAO8, PDO*** | 2.29 | 43.45 |
| NAO3, IOD1*** | 7.23 | 56.94 | NAO4, IOD7*** | 27.77 | 73.91 | NAO8, IOD5*** | 1.58 | 32.81 |
| NAO3, SIOD*** | 7.37 | 59.12 | NAO4, SIOD*** | 8.47 | 48.77 | NAO8, SIOD*** | 0.62 | 5.71 |
| AMO, PDO2*** | **2.79** | **26.04** | AMO5, PDO3*** | 27.51 | 72.81 | AMO8, PDO*** | 2.63 | 39.61 |
| AMO, IOD1*** | 3.49 | 41.23 | AMO5, IOD7*** | 22.08 | 71.07 | AMO8, IOD5*** | 1.45 | 33.23 |
| AMO, SIOD** | 3.51 | 42.22 | **AMO5, SIOD**\*** | **6.89** | **41.59** | **AMO8, SIOD**\*** | **0.17** | **10.21** |
| PDO2,IOD1** | 3.61 | 44.79 | PDO3, IOD7*** | 22.69 | 70.15 | PDO, IOD5 | 3.82 | 51.24 |
| PDO2, SIOD*** | 6.38 | 57.18 | PDO3, SIOD*** | 10.11 | 53.71 | PDO, SIOD | 2.23 | 43.02 |
| IOD1, SIOD*** | 5.49 | 51.22 | IOD7, SIOD*** | 7.35 | 43.02 | IOD5, SIOD | 2.44 | 43.78 |

Significance levels:

*** 0.001,

** 0.01,

* 0.05,

~ 0.1.

NAO–North Atlantic oscillation; AMO–Atlantic multidecadal oscillation; PDO–Pacific decadal oscillation, IOD–Indian Ocean dipole; SIOD–Sub-tropical Indian ocean dipole

regular fluctuations and multiple stages. Remote areas may simultaneously experience contrasting effects. During phase shifts, these regions experience contrasting yet altered effects. Oscillations and their anomalies have a major effect on marine ecosystems and particularly affect the movement and population dynamics of top predators such as tuna and marlins. For example, long-term climate variability and its influence on marine environmental conditions may alter the global population and distribution of yellowfin tuna (YFT). Decadal and multi-decadal climate fluctuations exhibit alternating effects across oceans at a frequency of approximately 8–16 years for YFT [55]. The NAO likely influences fisheries by inducing environmental and oceanic changes that either directly affect fish biology and fishing yields or indirectly trigger ecological cascades through fundamental changes in ecosystems, including changes to primary production [10]. Multidecadal climate variability, influenced by climate change, may affect the distribution of tuna and billfish in the NA basin [55]. This effect was also documented in other large pelagic species in the PO [56]. Climatic phenomena influence various aspects of the AO ecology, including ecosystem function, fishing resources, and carbon sequestration. This study's results are consistent with the hypothesis that variations in the catch rates for striped, blue, and silver marlins in the AO are linked to climatic oscillations. The catch rates for these three marlin species were found to be correlated with oscillations after certain delays, indicating a recurrent pattern.

## Effects of climatic oscillations on marlin biology

This study discovered that climatic oscillations had major delayed effects on the catch rates for the three marlin species, indicating a time lag between a climatic oscillation and its effect on fish populations (S3 Fig). Analysis of the delayed effects of five climatic oscillations revealed positive or negative correlations with the marlin catch rates. Various factors may contribute to the delayed effects of ocean conditions or climate cycles. For example, in response to gradual shifts in ocean conditions, fish populations may either acclimate or adapt. Acclimation refers to short-term modifications by individual species, whereas adaptation refers to long-term genetic modifications within populations. These two processes can be lengthy, leading to delayed shifts in distributions [57]. Changes in ocean conditions, such as vegetation or the depth and structure of the water column, affect the availability of suitable habitats for fish. Fish may require time to explore and adjust to their new habitat, resulting in a delay in their distributional response [58]. Overall, our findings regarding the delayed effects of climatic oscillations on catch rates are supported by those of a study on Pacific billfish [59]. This study noted that the effects of climate-scale oscillations, on the presence of billfish may be delayed over time. Another study shown that the consequences of delayed climatic oscillations had a greater effect on the Indian Ocean marlin species compared to immediate impacts [8]. Therefore, our current findings about the delayed effects of climatic oscillations, which are more significant than immediate impacts, are consistent with previous research. Long-lived species such as marlins are expected to take longer to respond to climatic oscillations compared with smaller pelagic fish, such as sardines and anchovies, which have shorter lifespans and thus respond with minimal lag. Teixeira et al. (2016) [60] discovered that the effect of the NAO on the recruitment of European sardines was delayed by less than 1 year. Faillettaz et al. (2019) [7] examined the effects of the 16-year-lagged AMO on the recruitment of long-lived BFT. Báez et al. (2020) [35] discovered that the distribution of YFT in the IO was significantly influenced by the delayed effects of climate cycles; the IOD, PDO, and MJO were found to be most strongly correlated with the YFT catch rate after delays of 6, 5, and 5 years, respectively. These variations in the lagged effect on YFT indicate the sensitivity of different fish species to ocean conditions. Certain species may respond rapidly to a shift in their environment. Overall, the

aforementioned studies lend support to our findings, underscoring the importance of delayed effects on marlin catch rates in the AO. Notably, marlin catch rates or distributions may not immediately change with the onset of a climatic oscillation, but they do shift after a delay as the fishes adapt to new ocean conditions.

## North Atlantic Oscillation

The NAO represents the exchange of atmospheric mass between the Arctic and subtropical Atlantic regions and affects weather conditions such as surface temperature, wind patterns, storm intensity, and precipitation in the NA. A positive NAO index is associated with intensified westerly winds at midlatitudes, originating from low-pressure anomalies in the Arctic and the Icelandic region and high-pressure anomalies in the subtropical Atlantic. This phenomenon leads to colder and drier conditions than average over the northwestern Atlantic and Mediterranean regions, with northern Europe, the eastern United States, and parts of Scandinavia experiencing weather that is warmer and wetter than average [10]. Several fishery resources are influenced by the NAO [16], resulting in a significant correlation between climatic patterns and fishery production. The NAO may also affect species' ecological traits of that influence fishing activities and yields, including recruitment, abundance, catchability, migration patterns, and fitness (physical condition), with varying effects on the biomass of fishing stock over time. This effect is both cumulative and dynamic, as indicated by Báez et al. (2021) [10], with a delay expected for certain outcomes, such as recruitment. In addition, the NAO exhibits a fluctuating pattern, with its effects varying in positive and negative phases, depending on the region affected by the phenomenon. For instance, an increase in primary output in the Alboran Sea as a result of a negative NAO phase may enhance the fitness of a single stock while having a detrimental effect on another stock in the North Sea, and *vice versa*. Therefore, different stocks may experience contrasting effects of the NAO, depending on their biogeographic setting. An illustrative example of the effect of the NAO on recruitment and fishery yields is the case of the Atlantic cod (*Gadus morhua* Linnaeus, 1758) [61]. Faillettaz et al. (2019) [7] identified a delayed impact of 16 years on the recruitment of BFT; they observed that adult BFT were abundant during both strongly negative and strongly positive NAO phases but that they were less abundant during periods with intermediate NAO values. The most adverse NAO phases negatively affected the recruitment of BFT, resulting in low abundance 16 years later. This phenomenon may explain the inconsistent findings that are often reported regarding the effect of the NAO on the recruitment and abundance of pelagic predators. For long-lived fish such as marlins, a significant delay exists between the influence of the NAO on the ecosystem and the resultant effects on fishery yields. Because of the effect of the NAO on recruitment, resource quantities may fluctuate [10]. This study discovered that a negative NAO phase coincided with low catch rates for striped and silver marlins but high catch rates for blue marlins (S1 Fig). These high catch rates for blue marlins during the negative NAO phase were attributable to an increase in food availability, such as zooplankton, influenced by the NAO. In several locations of the NA Ocean basin, the NAO affects the interannual variation in the abundance or biomass of zooplankton. In this region, increased food availability may facilitate the migration of fish schools and influence the biology of fish populations. Notably, the body condition of apex predators such as marlins is affected by fluctuations in food supply driven by the NAO. Migratory animals tend to have low energy requirements if the wind-driven surface currents are favorable during their migration in the NAO phase, resulting in an improvement in their body condition [62]. Enhanced physical fitness may improve spawning, leading to higher-quality eggs and increased offspring survival [10]. Understanding delays in ecosystems and their potential implications for fishery yields may aid

in mitigating negative outcomes. The NAO may affect the SST and thermocline, influencing fish catchability and local yield as a result of decreased fishing activity on stormy days and changes in fishery harvest [26].

## Atlantic Multidecadal Oscillation

Alheit et al. (2014) [12] discovered that the AMO affected the behavior of small pelagic fish and the occurrence of ecosystem regime shifts in the eastern NA and CA. Faillettaz et al. (2019) [7] reported that the AMO significantly affected the spatial distribution and regional abundance of BFT in the NA. In the present study, high blue marlin abundance was found to be associated with the 5-year-lagged AMO, whereas low striped and silver marlin abundance was associated with 0- and 8-year-lagged AMO, respectively. With a time lag of 5 years, the AMO remained the primary hydroclimatic factor for predicting the catch rates for blue marlins among the three marlin species, with the highest explained deviance (96.6%, Table 2). Generally, marlins are capable of traversing large distances quickly, which enables them to rapidly adjust to environmental shifts and migrate to the most favorable regions in the Atlantic. The productivity of these areas is predominantly affected by the AMO. Overall, our findings indicate that variations in the AMO may explain the oscillations in the recruitment and older life stages of marlins in the Atlantic. Our analysis also emphasizes the effect of the AMO on the basin-scale distribution of marlins. Specifically, the catch rates for blue marlins were found to be high only during the two favorable phases of the AMO from 2007 and 2009 onwards but started to decrease in subsequent years, indicating a strong lagged effect of the oscillation on catch rates. Wu et al. (2020) [55] reported that the AMO influenced the abundance of YFT in both the AO and globally, with a cycle of 8–16 years, confirming that the episodes of the AMO affect fishing vessel dynamics and species habitat suitability in the AO. In the present study, wavelet analysis revealed a positive cycle of 8–16 years with the AMO. Nevertheless, this analysis unveiled divergent patterns in the distributions of the three marlin species, as well as the influence of climate oscillations on various life phases. Since this was the preliminary investigation, the authors solely focused on identifying the delayed effects of climatic oscillations. In the subsequent phase, authors will endeavours to determine the influence of climatic oscillations and their delays on various life phases of marlin species, given that the findings of the preceding investigation have corroborated the current hypothesis. The authors planned to study the relationship between climatic oscillations and marlin life phases using a multi-faceted approach. They will collect extensive data on climatic patterns, and cross-reference it with historical and contemporary biological records of marlin species. They will use advanced statistical models and machine learning algorithms to analyze the correlations between climatic oscillations and marlin life phases, identifying potential lag effects and understanding the resilience and adaptability of marlin species to changing environmental conditions. The study will also include a geographical analysis to identify regional variations in the impact of climatic oscillations. The findings are expected to have significant implications for marine conservation strategies, helping to predict and mitigate climate change impacts on marlin populations and ensuring their sustainability for future generations.

## Pacific Decadal Oscillation

Overall, the correlation between the PDO and the AO affects marlin catch rates, indicating the interconnectedness of global climate patterns and marine ecosystems. Large pelagic fish such as billfish, tuna, and sharks experience major environmental shifts throughout their life cycles because of their extensive migration. Therefore, large-scale climatic indices that combine various physical characteristics as a proxy can be used for effectively predicting long-term

fluctuations in the abundance of these species. In the context of global climate change, understanding the relationship between ocean and atmosphere teleconnection patterns is essential for predicting the effects of climate patterns on the migration of large-scale species across different basins. Baez et al. (2020) [35] reported that the interaction between the PDO and SIOD had a delayed effect on the catch rates of YFT in the IO. This delayed effect was associated with increased recruitment, enhanced larval survival, and improved spawning conditions for YFT. According to observations, a negative PDO phase or a positive SIOD phase may increase fish stock abundance within 3–6 years, whereas a positive PDO phase or a negative SIOD phase may decrease fish stock abundance within the same time frame. Marlins presumably experience similar effects because of the characteristics they share with YFT, such as being a large pelagic predatory fish species with extensive migration. The PDO affects the global SST and the population and biomass of fish [63]. It also significantly affects the MLD and net primary productivity across the PO and AO.

## Indian Ocean Dipole

The tropical PO and AO exhibit a warm west–cold SST gradient because of the forcing of easterly trade winds. However, this gradient can weaken or reverse, which is accompanied by a slackening of trade winds. This phenomenon, referred to as El Niño, represents the positive phase of the ENSO in the Pacific and Atlantic Niño in the Atlantic [20]. In the tropical IO, the IOD is the primary mode of interannual climate variability that can influence the Atlantic Niño through atmospheric connections, even when it is not directly linked to the ENSO. A positive IOD phase typically leads to increased rainfall in the western tropical IO, resulting in anomalous westerly winds over the tropical AO and a weakening of the easterly trade winds in that region. These westward wind anomalies trigger oceanic downwelling Kelvin waves, induce abnormal eastward ocean circulation, and suppress oceanic upwelling, leading to warm anomalies in the equatorial AO [20]. This mechanism may explain the reduced marlin catch rates during positive IOD phases. Because of its seasonal pattern, the IOD has a major effect on the AO, with the IOD typically developing in the summer and peaking in the fall. During the summer, the average SST in the western tropical IO decreases as a result of the Indian summer monsoon. However, it then increases in the fall and winter, presumably contributing to the effect of tropical IO SST anomalies on local rainfall patterns. This phenomenon has a major influence on the tropical AO through atmospheric teleconnections.

## Subtropical Indian Ocean Dipole

Subtropical dipole events are the main mechanism underlying SST variability in the southern subtropical IO and AO, accounting for 27% of the SST variability observed in these regions. The SIOD plays a key role in triggering the tropical IOD [21]. Nevertheless, our understanding of the effect of SIOD on fish catches remains less comprehensive than our understanding of the influence exerted by other climatic patterns, such as the IOD and NAO. Anila and Gnanaseelan (2023) [22] outlined a cyclical feedback loop between the IOD and SIOD, particularly in tropical and subtropical areas and especially when the influence of the ENSO is minimal or absent. They indicated that, during this feedback cycle, the existence of a positive or negative SIOD phase tends to reinforce a positive or negative IOD phase, respectively, whereas the existence of a positive or negative IOD phase tends to reinforce a negative or positive SIOD phase, respectively. In the present study, the peak catch rate for striped marlins was discovered to be affected by a positive SIOD phase and occurred during a positive IOD event, which is typically associated with decreased catches. Similar results were obtained for blue and silver marlins,

wherein a positive SIOD phase coincided with a positive IOD phase, leading to reduced catch rates.

## Interactions of combined climatic patterns

An important discovery of this study is that the combined impact of several global climatic cycles is the most effective explanation for the fluctuations in marlin populations. Comparable correlations have been noted in several locations across the globe [64, 65]. It was observed that PDO lagged by 2 years together with AMO without any delay displayed the lowest GCV and AIC values revealing its degree of influence on striped marlin (Table 3). Since the 21st century, research has shown that AMO has a dominant influence on the long-term variations in the AO. This influence extends to other ocean basins, particularly the Pacific Ocean, where it affects prominent modes such as ENSO and PDO. The AMO also has a synchronous impact on the physical environment, leading to anomalous warm SST in the western Pacific, increased heat content in the upper ocean, and intensified tropical cyclones during positive AMO phases. These findings have been supported by various studies [66–69]. AMO was present for all the three species when the joint effect of oscillations was considered. This result suggests that the recruitment process of marlins may be influenced by the AMO, based on the maturity age in the AO and the observed causal association between the AMO and marlins with a lag varying from 0–8 years for the striped, blue and silver marlin respectively (Tables 2 and 3). During their journey of attaining maturity, they encounter the lagged variations in SST and other environmental parameters generated by AMO. The interactions between AMO and marlins exhibited time-dependent nonstationary responses, suggesting a potentially intricate ecological impact on the ecosystem. On the other hand, SIOD without any lag combined with AMO had the lowest GCV for both blue and silver marlins, indicating that SIOD tends to have an immediate effect on these two species rather than showing a delayed effect. These findings are crucial for making independent decisions regarding the marlin fishery in the AO.

Alheit et al. (2014) [12] showed that sardine and anchovy from the European Atlantic and Mediterranean coasts are influenced by AMO in the long run. Báez et al. (2022) [69] discovered that in the short-term, there is a positive correlation between sardine landings and AMO. Additionally, they discovered significant correlations between the amount of sardine population and AMO, as well as between the amount of anchovy population and physical condition. According to Alheit et al. (2014) [12], there has been a noticeable rise in the number and distribution of European anchovy in the North Sea since the mid-1990s linked to AMO. On the other hand, there has been a significant decline in sardine landings in the Western Mediterranean Sea from 2012 to 2022 [69]. In the long run, our results were comparable to the findings of Alheit et al. (2014) [12] for marlins from the AO. Nevertheless, we have discovered a persistent correlation between all three marlin species and AMO, either together with PDO or with SIOD as evident from the combined modelled effect. Due to the limited time frame of these findings, it is advisable to exercise caution when interpreting them. It is necessary to do additional study utilizing longer data series to obtain more reliable results. However, it is important to note that this time period is not uniform in terms of fishing characteristics due to notable advancements in technology and changes in the catch composition of the fleet, which may potentially affect the accuracy of the data.

It has been previously documented that the response of ecosystems can be better explained by distant climatic oscillations rather than by local physical variables, which is considered a paradox [10]. Coll et al. (2019) [70] suggest that recent prominent changes in fish populations may stem from various factors, such as the combined impact of environmental change, excessive fishing, competition for food resources, and predation. Nevertheless, it is important to

emphasize that favorable periods of climatic oscillations might greatly account for the patterns observed in marlin catch in the present study.

## Repercussions for fishery management as an implementation

The catch rate, a time series, may seem separate from physical phenomena like the climatic oscillations. However, they are interconnected due to their significant impact on marine ecosystems [55]. Changes in sea surface temperature, salinity, and ocean currents alter habitat conditions, affecting species distribution, breeding, feeding, and migration patterns. Changes in water temperature and nutrient availability can affect fish populations, affecting catch rates [45]. Additionally, climate oscillations, influenced by atmospheric and ocean interactions, lead to weather changes and oceanographic conditions, which are highly influenced by fish populations [43]. Integrating fishery time series data with climate oscillations provides a robust framework for understanding and predicting these interactions. Wavelet analysis is beneficial for analyzing simple time series fishery data due to its time-frequency localization, multi-resolution analysis, handling noise and non-stationarity, and its ability to compare time series like environmental indices and biological populations [48]. It allows for the identification of transient features, patterns, and structures, providing deeper insights into underlying processes [49]. Wavelet coherence aids in comparing time series like fishery and climatic oscillations data, providing a nuanced understanding of evolution over time [55].

As shown in Fig 3, the annual climate fluctuations caused by climatic oscillations in the AO may significantly affect fishery management relating to marlins. During adverse climate phases, input-based control measures are preferable to total allowable catch (TAC) strategies. As a result of oscillation-related variations, pelagic species regulated through vessel- and year-based catch quotas may experience years of high recruitment and years of low recruitment. Regardless of stock abundance and recruitment levels, fishers typically strive to achieve the maximum authorized quota, potentially causing disproportionate harm in adverse years. Effort-based management strategies, such as setting a maximum number of fishing days, can

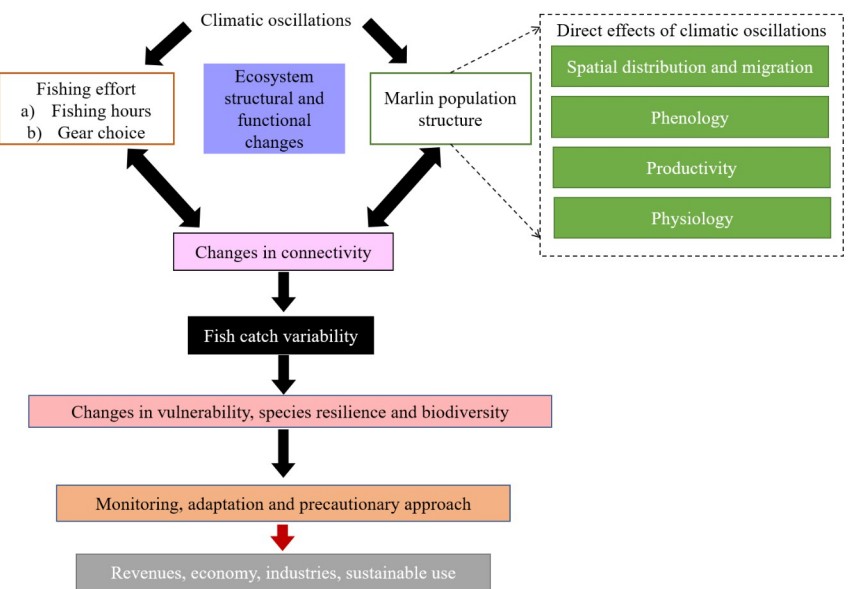

**Fig 3. The relationship between climatic oscillations and marlin catch rates through ecological (oceanographic changes) and behavioral (fisher behavior) processes targeting monitoring and adaptation.**

aid in synchronizing catches with relative abundance in a given year. During years with favorable oscillations and high recruitment, fisherman may benefit from increased yields per set, with catch rates being lower during unfavorable seasons. Unfavorable years adversely affect the fitness of large, slow-growing fish such as marlins, resulting in lower spawning success and recruitment rates. Therefore, implementing a fixed TAC strategy may increase fishing mortality, with fisheries operating under input-based control presumably adopting seasonal closures. During environmental changes, fishery control based on fishing effort may be suitable if these changes are effectively monitored. Overall, our findings regarding the relationship between marlin catch rates and climatic oscillations provide valuable insights for the management of marlin species, particularly through the adoption of early warning systems. Severe weather events may have a major effect on marine ecosystems and the availability of fish stocks, indicating the importance of early warning systems. Such essential information may aid governments in devising and implementing strategies aimed at minimizing negative effects on the fishing industry and coastal communities, thereby enhancing the resilience of marine ecosystems and improving socioeconomic outcomes.

## Conclusion

Climate change is expected to significantly impact marine ecosystems, particularly the Atlantic, affecting highly migratory species like marlins. These changes will affect their biology, ecology, distribution, abundance, survival, reproductive success, and management. Understanding and managing these changes is crucial for the future presence and abundance of HMS, as climate change and interdecadal climate variability will influence oceanographic processes. This study found that climatic oscillations significantly impact the catch rates of striped, blue, and silver marlins in the Atlantic Ocean (AO), with different lag times and specific climate indices influencing each species uniquely. Notably, the annual SIOD index strongly correlated with striped marlin catch rates, the AMO index lagged by 5 years with blue marlin, and the SIOD with silver marlin. The results highlight the complexity of marlin population dynamics and the combined effects of climate oscillations and overfishing, underscoring the need for further research on population dynamics and incorporating fishing data to better understand and mitigate these impacts. Overall, our findings suggest that marine scientists, policymakers, and coastal communities must acknowledge the influence of climatic oscillations on ocean conditions. Further research is required to enhance the management of marine resources, improve predictions of changes in ocean ecosystems, and facilitate the development of strategies aimed at adapting to and mitigating the effects of climatic variability on the oceans. Such insights may extend to marine biodiversity, coastal economies, and the overall health of the climate system.

## Supporting information

**S1 Table. Sources of various climate oscillation data.**
(DOCX)

**S2 Table. Evaluating the influence of climatic fluctuations on the catch rate of three marlin species with Generalized Additive Models (GAMs).** The selected items are the ones in bold.
(DOCX)

**S1 Fig. Partial effect plots of the marlins with the most significant lag of each climatic oscillation.**
(DOCX)

**S2 Fig. VIF analysis results for the effect of climatic oscillations on marlin catch rates.** The dashed red line represents the VIF threshold.
(DOCX)

**S3 Fig. Annual fluctuations in the capture of three marlin species over the research period with the chosen lag for each climatic oscillation.**
(DOCX)

## Author Contributions

**Conceptualization:** Sandipan Mondal, Ming-An Lee.

**Data curation:** Sandipan Mondal, Malagat Boas.

**Formal analysis:** Sandipan Mondal.

**Funding acquisition:** Ming-An Lee.

**Investigation:** Ming-An Lee, Subhadip Dey.

**Methodology:** Sandipan Mondal.

**Project administration:** Ming-An Lee.

**Resources:** Ming-An Lee.

**Software:** Sandipan Mondal, Malagat Boas, Sawai Navus, Koushik Kanti Barman.

**Supervision:** Ming-An Lee, Subhadip Dey.

**Validation:** Aratrika Ray, Ming-An Lee, Subhadip Dey.

**Visualization:** Aratrika Ray, Sawai Navus, Ming-An Lee.

**Writing – original draft:** Sandipan Mondal, Aratrika Ray.

**Writing – review & editing:** Sandipan Mondal, Aratrika Ray, Koushik Kanti Barman.

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
