## [Decision Letter · Decision Letter 0]

29 May 2024

PONE-D-24-14238Can the delayed effects of climatic oscillations have a greater influence on global fisheries compared to their immediate effects?PLOS ONE

Dear Dr. Lee,

Thank you for submitting your manuscript to PLOS ONE. After careful consideration, we feel that it has merit but does not fully meet PLOS ONE’s publication criteria as it currently stands. Therefore, we invite you to submit a revised version of the manuscript that addresses the points raised during the review process.

We look forward to receiving your revised manuscript.

Kind regards,

Claudio D'Iglio, Ph.D.

Academic Editor

PLOS ONE

Journal Requirements:

"National Science & Technology Council of Taiwan (NSTC), NSTC 112-2811-M-019-004. "

"We are extremely thankful to the National Science & Technology Council of Taiwan (NSTC) for funding this research under the grant number 112-2811-M-019-004. "

"National Science & Technology Council of Taiwan (NSTC), NSTC 112-2811-M-019-004. "

"The authors declare that they have no competing interests."

6. We note that you have indicated that there are restrictions to data sharing for this study. PLOS only allows data to be available upon request if there are legal or ethical restrictions on sharing data publicly. For more information on unacceptable data access restrictions, please see http://journals.plos.org/plosone/s/data-availability#loc-unacceptable-data-access-restrictions. 

Additional Editor Comments:

Dear Dr. Sandipan Mondal,

Thank you for submitting your manuscript to PLOS ONE. After careful consideration, we feel that it has merit but does not fully meet PLOS ONE’s publication criteria as it currently stands. Therefore, we invite you to submit a revised version of the manuscript that addresses the points raised during the review process.

Reviewers' comments:

Reviewer's Responses to Questions

**Comments to the Author**

1. Is the manuscript technically sound, and do the data support the conclusions?

Reviewer #1: Partly

Reviewer #2: Yes

2. Has the statistical analysis been performed appropriately and rigorously? 

Reviewer #1: No

Reviewer #2: Yes

3. Have the authors made all data underlying the findings in their manuscript fully available?

Reviewer #1: No

Reviewer #2: Yes

4. Is the manuscript presented in an intelligible fashion and written in standard English?

Reviewer #1: No

Reviewer #2: Yes

5. Review Comments to the Author

Reviewer #1: Teleconnections between oscillations and fishery yield is not new .. what exactly the novelty the authors tries to bring is to be elaborated

L96 onwards must be redrafted ..her it is the summary of work done /future studies etc .. but what expected is the research gap and stating the objectives precisely.. say (i) .. (ii)

L146 and L192 what is the difference ? here wavelet coherence is used in the latter case . how it is different from correlation ? why you need to use coherence ?

Basically wavelet coherence is used to find association between catch rate and oscillation .. what is the logic in taking the series of catch rate and its wavelet analysis ?how you physically justify it ? oscillation is a process /physical phenomenon

Also some more details on wavelet implementation is required if the above one is properly justified

Other than being two time series, how you physically justify using the analysis for catch rate ?..the relation is more statistical than physical ?

How the joint effect of multiple oscillations influence the results ? to be analysed and explained

Conclusion is more generic .. quantitative information should be added and revised

Reviewer #2: The study PONE-D-24-14238 provides new rigorous information regarding the ralationship between climatic ocillations and the catch rate of three commercially important marlin species. The MS is in my opinion rather sound with a solid statistical analysis which sustains the discussion. The text in my opiinion is clear and mostly easy to be followed, but sinceI am not a native speaker, I will refrain from commenting on its grammar or style. On the contrary, I admire the writers for presenting the vast number of results (and acronyms) in a way that did not result in a way that didn't ended in a confusing or even dull text.

Nonetheless, I do think that the MS still present some imprecisions and and other small things that need to be amended before its pubblication.

Therefore, I’d endorse the publication of the manuscript after some MINOR revisions.

For authors convenience a list of my minor comments and suggestions are provided in the attached pdf file.

I remain

6. PLOS authors have the option to publish the peer review history of their article (what does this mean?). If published, this will include your full peer review and any attached files.

Reviewer #1: No

Reviewer #2: No

---

## [Author Response · Author response to Decision Letter 0]

17 Jun 2024

Comment-wise Replies to Reviewer 1: 

1. Teleconnections between oscillations and fishery yield is not new .. what exactly the novelty the authors tries to bring is to be elaborated We thank the reviewer for the comment.

Ans. Climatic oscillations through teleconnections present a favorable occasion to examine and assess the impact of microhabitat changes on the distribution of fish. Despite our extensive knowledge, significant gaps still exist, especially regarding the consequences of climate change on the intricate food web that sustains marlin populations and their fisheries. It is uncertain whether climate change will affect the relationships between marlin fishing and the structure and functioning of the Atlantic Ocean ecosystem. By identifying changes in the distribution of marlin, we can enhance our comprehension of the habitat needs of crucial marine resources and improve our future management of marlin fisheries. There is a lack of comprehensive data on climate variability and its effects on marlin resources in the Atlantic Ocean. Undoubtedly, comprehending the impact of the oceanic environment on the distribution of tuna species is a crucial prerequisite for implementing ecosystem-based management of fisheries, which is progressively becoming a fundamental necessity in management policy. Therefore, the novelty of this research lies in the fact that it has tried to investigate the impact of climatic oscillations on marlin availability in the Atlantic Ocean. There has been no previous research that has examined this aspect and our study is the first one to also find the association of three marlin species whose distribution or catch rate was found to be influenced by three oscillations despite being located in the same ocean, that also points to the species-specific responses to climatic changes. 

2. L96 onwards must be redrafted ..her it is the summary of work done /future studies etc .. but what expected is the research gap and stating the objectives precisely.. say (i) .. (ii) 

Ans. The section has been re-drafted as suggested by the reviewer and the research objectives and gaps have been addressed now.

Line 128 – 150 (File name: Manuscript)

3. L146 and L192 what is the difference ? here wavelet coherence is used in the latter case . how it is different from correlation ? why you need to use coherence ? 

Ans.We thank the reviewer for the comment. Wavelet analysis is a widely employed technique for analyzing time series data. It is a method that enables analysis of periodic components and their temporal variations by representing the data in a time–frequency space. The technique computes a wavelet coefficient for each location in the dataset by measuring the correlation between the observed signal (in this case, the selected climatic index) and the wavelet. The outcome is a matrix containing wavelet coefficients for every possible combination of scale (representing the frequency of the wavelet) and translation (indicating the shift of the wavelet along the signal). Wavelets provide the advantage of simultaneously examining many scales. The wavelet transform breaks down a time series into its constituent time, frequency, and power components, which can be seen in a three-dimensional space using the wavelet power spectrum (WPS) plot. In a WPS plot, the x-axis represents the time series, while the y-axis represents the contribution of frequencies, which is indicated by the term "period". The term "power" refers to the measure of the amount of variation present in a time series at a specific wavelet. The WPS assesses the aspects of the signal that are influential and contributory, as well as those that are less important. Wavelet coherence was used to investigate environmental variables with the greatest predictive power for analysis of fisheries productivity. The wavelet coherence was the mean coherence within the significant coherence zone, so a larger coherence indicated a more significant contribution of the variable to explain variability in response variables. These plots were used to identify periods of greatest correlation (represented by red points in the figures). On the other hand, one of the most commonly used methodologies to identify potential relationships between variables in climate research is correlation, with or without a lag (or time delay). Such correlation (or linear regression) approaches, despite being useful for identifying potential relationships between variables, do not imply causation. A significant correlation simply means that there is a relationship, or synchronous behavior, between two variables without explicitly confirming a causal link between the two. Thus, as a preliminary step, the cross-correlation between different variables and marlin catch rate was computed in the present study. The same methodology has been followed by Adarsh et al., 2024. Wavelet coherence combines the advantages of wavelet analysis and Pearson correlation, allowing for searching for correlations that vary over frequency and time. 

Ref. Adarsh, S., Fathima, S. and Arunkumar, R., 2024. Multiscale teleconnection analysis of rainfall patterns over Calicut, India using wavelet coherence. Journal of Earth System Science, 133(1), pp.1-16.

Ma, S., Huse, G., Ono, K., Nash, R.D., Sandø, A.B., Nedreaas, K., Sætre Hjøllo, S., Sundby, S., Clegg, T., Vølstad, J.H. and Kjesbu, O.S., 2024. Recruitment regime shifts and nonstationarity are widespread phenomena in harvestable stocks experiencing pronounced climate fluctuations. Fish and Fisheries, 25(2), pp.320-348.

Mathews, J. and Czaja, A., 2024. Oceanic maintenance of atmospheric blocking in wintertime in the North Atlantic. Climate Dynamics, pp.1-14.

4. Basically wavelet coherence is used to find association between catch rate and oscillation .. what is the logic in taking the series of catch rate and its wavelet analysis ?how you physically justify it ? oscillation is a process /physical phenomenon We thank the reviewer for the comment.

Ans. Wavelet analysis uses a set of functions locally defined in both time and frequency domains. Wavelet analysis performs a time-frequency analysis of the signal, which permits the estimation of the spectral characteristics of the signal as a function of time and then the identification of different periodic components and their time evolution all along the time series. In other words, transient dynamics or gradual changes of the periodic components of the signal can be detected. Furthermore, cross-wavelets and phase analysis generalize these possibilities to the analyses of dependencies between two signals. Wavelet coherence, however, can help resolve relationships when time lags confound correlation. Two variables are coherent if they exhibit correlated oscillations and consistent phase differences through time, even if the variables are phase-lagged or anti-phase (in our study these two variables are the marlin catch rate and oscillation). Consistency of phase differences is important because it is more likely to imply a causal relationship that is not due to chance. Furthermore, identification of the type of phase relationships can help inform the potential drivers. Like other wavelet approaches, wavelet coherence also accommodates complex temporal autocorrelation structures, another common property of fisheries time series that confound correlation tests. Therefore, we used wavelet coherence analyses to determine whether marine marlin fishery catches were coherent in the study region during the time considered, and to determine the distribution of phase relationships between fishery catches. 

Ref. Sanz-Fernández, V., Gutiérrez Estrada, J.C., Queirolo Palma, D. and Hormazábal Fritz, S., Climatic and Oceanographic Variability Associated with Historical Landings of Blackspot Seabream in the Strait of Gibraltar: Wavelet-Based Analysis. Available at SSRN 4721183.

Ong JJL, Walter JA, Jensen OP, Pinsky ML. Global hotspots of coherent marine fishery catches. Ecol Appl. 2021 Jul;31(5):e02321. 

5. Also some more details on wavelet implementation is required if the above one is properly justified We thank the reviewer for the comment.

Ans. More details on wavelet implementation have now been added in the revised manuscript like the responses above.

Line 229 – 258 

6. Other than being two time series, how you physically justify using the analysis for catch rate ?..the relation is more statistical than physical ? We thank the reviewer for the comment.

Ans. Understanding the relative effects of the array of factors affecting population dynamics is a primary objective of fisheries science. It is widely accepted that fishing activity modifies population demography and structure, stock sizes and trophic interactions. Climate induced variability is also a strong driver of change in fish populations. Research has focused on associations between climatic process, fishing resources and effects on the egg and larval phases when these early life stages of studied species are highly sensitive compared to later (juvenile and adult) life stages. The influence of climate variability on fish populations can be classified on three different types: direct effects on the survival of pre-recruits or larvae; indirect effects on spawning stock biomass (by changing seasonal reproduction cycle; and mixed effects – a combination of direct and indirect effects. Studies on climate driven effects over fisheries span over a century. 

Analysing time series through wavelet or prediction models have been employed to identify the potential cause of environmental impacts on marine fishing resources. The establishment of connections between environmental data and fish stock characteristics or parameters, as well as the integration of these findings into assessments, should be grounded in a comprehensive comprehension of the ecology of the system or region being examined. Furthermore, it should offer insight into the mechanisms elucidated by statistical correlations. The process of determining the causes of variations in landings is intricate, leading to numerous potential ideas to elucidate observed patterns. By employing various analytical methodologies or statistical models, researchers can generate many hypotheses to explain landing patterns. This can lead to a wide range of potential models, many of which may effectively fit the data. Hence, it is imperative to make progress in the assessment of statistical models by employing a model selection technique that emphasizes the evaluation of the models' "predictive" capacities rather than their performance in fitting the data.

Multiple authors concur that marine species have the potential to adapt to climate oscillations by altering their phenology, abundance, distribution, and recruitment. Several studies have examined how climatic cycles affect the abundance and captures of species that are the focus of fisheries. Hence, it is crucial to assess if climate oscillations lead to variations in the abundance and catchability of commercially valuable species, thereby influencing fish market prices by affecting the supply of fishing goods, that can be only done when the catch rate is analysed in junction with the ambient environment of the fish. 

Ref. Baptista, V. and Leitão, F., 2014. Commercial catch rates of the clam Spisula solida reflect local environmental coastal conditions. Journal of Marine Systems, 130, pp.79-89.

Vânia, B., Ullah, H., Teixeira, C.M., Range, P., Erzini, K. and Leitão, F., 2014. Influence of environmental variables and fishing pressure on bivalve fisheries in an inshore lagoon and adjacent nearshore coastal area. Estuaries and coasts, 37, pp.191-205.

7.How the joint effect of multiple oscillations influence the results ? to be analyzed and explained We thank the reviewer for the comment.

Ans. The joint effect of pairs of climatic oscillations has been analysed (Table 3) and explained in detail as per the suggestions.

Line 581 – 621 

8.Conclusion is more generic .. quantitative information should be added and revised We thank the reviewer for the comment.

Ans.The conclusion has been revised and restructured as per the suggestions. 

Line 647 – 694 

Comment-wise Replies to Reviewer 2: 

1. Line 17-18: This part is unclear. in the first lines of the abstract it has been stated that the study will examine catch rates in the Atlantic Ocean, while here are reported results from Indian Ocean. Please consider modify either this sentence or the previous ones. 

Ans. We would like to thank the reviewer for valuable comments. Various climate oscillations from different oceans have an impact on the Atlantic Ocean through teleconnection. Teleconnection is a climate phenomenon where weather patterns in one region are influenced by conditions in distant ones, facilitated by large-scale atmospheric or oceanic processes.

For instance, the Indian Ocean dipole (IOD) impacts atmospheric circulation patterns such as the Walker and Hadley circulations, affecting global weather conditions, including Atlantic hurricane development. Changes in sea surface temperatures propagate through ocean currents, indirectly affecting Atlantic region weather patterns and climate phenomena.

The Subtropical Indian Ocean Dipole (SIOD) significantly impacts the Atlantic Ocean through variations in sea surface temperatures, affecting atmospheric circulation patterns, Walker circulation, and precipitation patterns. These anomalies can affect tropical cyclone development, affecting the Atlantic basin and surrounding areas, and playing a role in global climate systems beyond the Indian Ocean.

Ref: An S-I, Wang C, Mechoso CR. Teleconnections in the Atmosphere. In: Mechoso CR, ed. Interacting Climates of Ocean Basins: Observations, Mechanisms, Predictability, and Impacts. Cambridge University Press; 2020:54-88.

Prior research has also demonstrated the significance of teleconnections in numerous cases. 

Ref: Zhang, L. and Han, W., 2021. Indian ocean dipole leads to Atlantic Niño. Nature Communications, 12(1), p.5952.

Báez, J.C., Czerwinski, I.A. and Ramos, M.L., 2020. Climatic oscillations effect on the yellowfin tuna (Thunnus albacares) Spanish captures in the Indian Ocean. Fisheries Oceanography, 29(6), pp.572-583.

The original study focused exclusively on marlin species found in the Atlantic Ocean. Nevertheless, due to the aforementioned reasons, authors also included the use of IOD and SIOD to examine any potential influence of teleconnections on the catch rates of Atlantic Ocean marlin species.

2. Line 45: Revised as per the suggestion. Line 54 (File name: Manuscript)

3. Line 97: Please stated here which species and why have been selected

Ans. We would like to thank the reviewer for valuable comments. Three marlin species were included in the present study and are as follows:

 1. Striped marlin - Kajikia audax, 1887; 

 2. Blue marlin - Makaira nigricans, 1802; 

 3. Silver marlin - Kajikia albida, 1860

Two species reasons are there behind selecting these species only and are as follows:

 1. The specific rationale for choosing only these three marlin species was based on the fishery data only provided by the Overseas Fisheries Development Council of Taiwan, which exclusively consisted of data pertaining to these three marlin species.

 2. These three marlin species are vital species in the Indian Ocean, playing a crucial role as apex predators, regulating prey populations, supporting commercial and recreational fishing industries, and generating revenue through tourism. They also symbolize maritime heritage and local traditions. Line 112-120 (File name: Manuscript)

4. Line 107-109: This sentence seems bit awkward to me. Please consider rephrasing. 

Ans. We would like to thank the reviewer for valuable comments. Authors added some important information also, while rephrasing this line.

“Despite these benefits, no studies have yet comprehensively examined the effects of climatic oscillations and ocean conditions on the catch rates and distribution of marlins in the AO (Mondal et al., 2023). In addition, there are concerns about the stock condition of marlins, in the Atlantic Ocean, as there are indications of over

---

## [Decision Letter · Decision Letter 1]

4 Jul 2024

PONE-D-24-14238R1Can the delayed effects of climatic oscillations have a greater influence on global fisheries compared to their immediate effects?PLOS ONE

Dear Dr. Lee,

Thank you for submitting your manuscript to PLOS ONE. After careful consideration, we feel that it has merit but does not fully meet PLOS ONE’s publication criteria as it currently stands. Therefore, we invite you to submit a revised version of the manuscript that addresses the points raised during the review process.

The MS has been strongly improved according to the reviewers suggestions. Here  the second round of revision is reported with some minor suggestions usefull to further improve the MS quality and clarity. I strongly suggest authors to modify and correct the MS according to the Reviewers suggestions.

We look forward to receiving your revised manuscript.

Kind regards,

Claudio D'Iglio, Ph.D.

Academic Editor

PLOS ONE

Journal Requirements:

Reviewers' comments:

Reviewer's Responses to Questions

**Comments to the Author**

1. If the authors have adequately addressed your comments raised in a previous round of review and you feel that this manuscript is now acceptable for publication, you may indicate that here to bypass the “Comments to the Author” section, enter your conflict of interest statement in the “Confidential to Editor” section, and submit your "Accept" recommendation.

Reviewer #1: All comments have been addressed

Reviewer #2: All comments have been addressed

2. Is the manuscript technically sound, and do the data support the conclusions?

Reviewer #1: Partly

Reviewer #2: Yes

3. Has the statistical analysis been performed appropriately and rigorously? 

Reviewer #1: N/A

Reviewer #2: Yes

4. Have the authors made all data underlying the findings in their manuscript fully available?

Reviewer #1: No

Reviewer #2: Yes

5. Is the manuscript presented in an intelligible fashion and written in standard English?

Reviewer #1: No

Reviewer #2: Yes

6. Review Comments to the Author

Reviewer #1: Authors made a good attempt to revise the manuscript..i have few minor points

1. The use of wavelets etc the justification should be supported by citing some recent references in the main text

2. Similarly, the wavelet application in this catches ..is it completely novel ? else to be supported with past studies in the main text

3. Qn 4 need more clarity .. catche rate is just a time series, while AMO etc are a physical process .. how the application of wavelet on a simple time series is justifiable ? for example if the catche series shows a periodicity of 30 .. what does it mean ?

4. Add unit to Period in all wavelet Figs .. also enlarge the font size

Reviewer #2: This revised version of the Manuscript PONE-D-24-14238R1 has made significant improvements. I want to express my gratitude to the authors for carefully considering all of my recommendations and remarks. The paper is now, in my opinion, practically ready for publication. As for the previous review process I've appended below a (brief) list of my minor suggestion. As a result, I propose accepting this manuscript with a few Minor revisions.

Lines141-142: Please also add the species descriptor

Lines 155-157 and 157-160: please add references

Lines 155-157: I’d suggest adding references here

Line 605: “vice versa” it should probably be in italic

Lines 663-666: I would suggest spending few more words on how the authors are planning to investigate these aspects

7. PLOS authors have the option to publish the peer review history of their article (what does this mean?). If published, this will include your full peer review and any attached files.

Reviewer #1: No

Reviewer #2: No

---

## [Author Response · Author response to Decision Letter 1]

7 Jul 2024

Comment-wise Replies to Reviewer 1: 

Comments Replies

1. The use of wavelets etc. the justification should be supported by citing some recent references in the main text We would like the thank the reviewer for this important comment. Following references are added now.

1. Lee, M.A., Mondal, S., Wu, J.H., Huang, Y.H. and Boas, M. Cyclic Variation in Fishing Catch Rates-Influenced by Climatic Variability in the Waters Around Taiwan. Journal of Taiwan Fisheries Society, 49(2), pp.113-125, 2022,

https://doi.org/10.29822/JFST.202206_49(2).0005. Ref 36.

2. Yadav, V.K., Jahageerdar, S. and Adinarayana, J. Modeling framework to study the influence of environmental variables for forecasting the quarterly landing of total fish catch and catch of small major pelagic fish of North-West Maharashtra coast of India. National Academy Science Letters, 43(6), pp.515-518, 2020,

https://doi.org/10.1007/s40009-020-00922-2. 

Ref 63. 

3. Raubenheimer, C. and Phiri, A. The impact of climate change and economic development on fisheries in South Africa: a wavelet-based spectral analysis. Humanities and Social Sciences Communications, 10(1), pp.1-11, 2023,

https://doi.org/10.1057/s41599-023-02408-0.

Ref 48.

4. Lan, K.W., Wu, Y.L., Chen, L.C., Naimullah, M. and Lin, T.H., 2021. Effects of climate change in marine ecosystems based on the spatiotemporal age structure of top predators: A case study of bigeye tuna in the Pacific Ocean. Frontiers in Marine Science, 8, p.614594, 2021,

https://doi.org/10.3389/fmars.2021.614594. 

Ref 33.

2. Similarly, the wavelet application in this catches ..is it completely novel ? else to be supported with past studies in the main text We would like the thank the reviewer for this important comment. Following references are added now.

1. Wu, Y.L., Lan, K.W., Evans, K., Chang, Y.J. and Chan, J.W. Effects of decadal climate variability on spatiotemporal distribution of Indo-Pacific yellowfin tuna population. Scientific Reports, 12(1), p.13715, 2022, 

https://doi.org/10.1038/s41598-022-17882-w. 

Ref 62.

2. Lan, K.W., Chang, Y.J. and Wu, Y.L. Influence of oceanographic and climatic variability on the catch rate of yellowfin tuna (Thunnus albacares) cohorts in the Indian Ocean. Deep Sea Research Part II: Topical Studies in Oceanography, 175, p.104681, 2020,

https://doi.org/10.1016/j.dsr2.2019.10468. 

Ref 34.

3. Qn 4 need more clarity .. catche rate is just a time series, while AMO etc are a physical process .. how the application of wavelet on a simple time series is justifiable ? for example if the catche series shows a periodicity of 30 .. what does it mean ?

“Basically wavelet coherence is used to find association between catch rate and oscillation .. what is the logic in taking the series of catch rate and its wavelet analysis ?how you physically justify it ? oscillation is a process /physical phenomenon” We would like the thank the reviewer for this important comment. 

(A) Catch rate is just a time series, while AMO etc. are a physical process, how you physically justify it ? oscillation is a process /physical phenomenon

Ans. The catch rate, while considered as a time series, may appear separate from physical phenomena such as the Atlantic Multidecadal Oscillation (AMO). However, there is a compelling rationale for examining them in conjunction, as environmental conditions, which are influenced by physical processes (climatic oscillations), can have a substantial effect on biological and ecological phenomena. Physical Justification can be as follows:

1. Environmental Influence on Marine Life: Any physical process like climatic oscillations involving changes in sea surface temperature, salinity, and ocean currents, impacts marine ecosystems by altering habitat conditions, affecting species distribution, breeding, feeding, and migration patterns. Changes in water temperature and nutrient availability can affect fish populations, affecting catch rates (Lee et al., 2022, Lan et al., 2020).

2. Climate-Ocean Interactions: Climate oscillations, are influenced by intricate interactions between the atmosphere and ocean, leading to weather changes and subsequently affecting oceanographic conditions. These changes are highly influenced by fish populations, affecting catch rates (Wu et al., 2022, Yadav et al., 2020).

In summary, the analysis of catch rates alongside physical processes like the AMO and other climatic oscillations is justified due to the significant impact of these oscillations on marine ecosystems. The integration of fishery time series data with climate oscillations through statistical and mechanistic approaches provides a robust framework for understanding and predicting these interactions.

Brief description is added to the discussion section as follows:

“The catch rate, a time series, may seem separate from physical phenomena like the Atlantic Multidecadal Oscillation (AMO). However, they are interconnected due to their significant impact on marine ecosystems. Changes in sea surface temperature, salinity, and ocean currents alter habitat conditions, affecting species distribution, breeding, feeding, and migration patterns. Changes in water temperature and nutrient availability can affect fish populations, affecting catch rates. Additionally, climate oscillations, influenced by atmospheric and ocean interactions, lead to weather changes and oceanographic conditions, which are highly influenced by fish populations. Integrating fishery time series data with climate oscillations provides a robust framework for understanding and predicting these interactions.” Line 796-805

(B) The application of wavelet analysis to a simple time series is justifiable for several reasons:

1. Wavelet transforms offer time-frequency localization, allowing analysis of signal frequency changes over time, particularly useful for non-stationary fishery time series data, identifying transient features (Raubenheimer & Phiri, 2023).

2. Wavelets enable multi-resolution analysis, allowing fishery time series data to be examined at different scales, identifying patterns and structures, and providing deeper insights into the underlying processes (Lan et al., 2021).

3. Wavelet transforms effectively handle noise and non-stationarity in real-world time series fishery data by identifying and filtering out noise, thereby improving the understanding of the underlying signal (Wu et al., 2022).

4. Wavelet coherence aids in comparing time series, like environmental indices and biological populations, by identifying time-localized correlation structures, providing a nuanced understanding of evolution over time (Yadav et al., 2020).

Brief description is added to the discussion section as follows:

“Wavelet analysis is beneficial for analyzing simple time series fishery data due to its time-frequency localization, multi-resolution analysis, handling noise and non-stationarity, and its ability to compare time series like environmental indices and biological populations. It allows for the identification of transient features, patterns, and structures, providing deeper insights into underlying processes. Wavelet coherence aids in comparing time series like fishery and climatic oscillations data, providing a nuanced understanding of evolution over time.” Line 805-813

4. Add unit to Period in all wavelet Figs .. also enlarge the font size We would like the thank the reviewer for this important comment. Figures are rectified now.

Comment-wise Replies to Reviewer 2: 

Comments Replies

Lines141-142: Please also add the species descriptor We would like the thank the reviewer for this important comment. Descriptor is now added in the revised manuscript as follows:

“The striped marlin, a renowned predator in recreational fishing, is known for its elongated body, vivid blue and silver hues, and large dorsal fin. Its vertical stripes become more prominent when stimulated or engaged in hunting. The blue marlin, with its vibrant cobalt blue upper body and shiny white underbelly, is a large predator with a long, pointed beak and a tall, curved dorsal fin. Its main food is fish and cephalopods. The silver marlin, with its streamlined, elongated body, cobalt blue dorsal surface, and pristine silvery ventral region, is a sought-after species for its dexterity and swiftness. The Atlantic white marlin's dorsal fin is conspicuous and curved, contributing to its sleek and streamlined appearance. The species primarily feeds on tiny fish and squid, making it an effective predator in its marine environment. These species are highly valued in recreational fishing for their dexterity and robustness.” Line 141-152.

Lines 155-157 and 157-160: please add references We would like the thank the reviewer for this important comment. Following references are added now.

1. Sorenson, L., McDowell, J.R., Knott, T. and Graves, J.E. Assignment test method using hypervariable markers for blue marlin (Makaira nigricans) stock identification. Conservation Genetics Resources, 5, pp.293-297, https://doi.org/10.1007/s12686-012-9747-x, 2013. 

Ref 53.

2. Conley, K.R. and Sutherland, K.R. Commercial fishers' perceptions of jellyfish interference in the Northern California Current. ICES Journal of Marine Science, 72(5), pp.1565-1575, https://doi.org/10.1093/icesjms/fsv007, 2015.

Ref 18.

3. Restrepo, V., Diaz, G. A., & Cortes, E. Stock assessment of Atlantic striped marlin (Kajikia audax) using catch data and abundance indices from commercial and recreational fisheries. ICES Journal of Marine Science, 68(9), 1884-1893. https://doi.org/10.1093/icesjms/fsp136, 2009. Ref 49.

4. Mamoozadeh, N. R., Graves, J. E., Bealey, R., Schratwieser, J., Holdsworth, J. C., Ortega-Garcia, S., & McDowell, J. R. Genomic data resolve long-standing uncertainty by distinguishing white marlin (Kajikia albida) and striped marlin (K. audax) as separate species. ICES Journal of Marine Science, 80(6), 1802-1813. https://doi.org 10.1093/icesjms/fsad136, 2023. Ref 39.

Lines 155-157: I’d suggest adding references here We would like the thank the reviewer for this important comment. Following references are added now.

1. Sorenson, L., McDowell, J.R., Knott, T. and Graves, J.E. Assignment test method using hypervariable markers for blue marlin (Makaira nigricans) stock identification. Conservation Genetics Resources, 5, pp.293-297, https://doi.org/10.1007/s12686-012-9747-x, 2013. 

Ref 53.

2. Conley, K.R. and Sutherland, K.R. Commercial fishers' perceptions of jellyfish interference in the Northern California Current. ICES Journal of Marine Science, 72(5), pp.1565-1575, https://doi.org/10.1093/icesjms/fsv007, 2015.

Ref 18.

Line 605: “vice versa” it should probably be in italic Rectified now. Line 606 

Lines 663-666: I would suggest spending few more words on how the authors are planning to investigate these aspects We would like the thank the reviewer for this important comment. More description is added now as follows:

“The authors planned to study the relationship between climatic oscillations and marlin life phases using a multi-faceted approach. They will collect extensive data on climatic patterns, and cross-reference it with historical and contemporary biological records of marlin species. They will use advanced statistical models and machine learning algorithms to analyze the correlations between climatic oscillations and marlin life phases, identifying potential lag effects and understanding the resilience and adaptability of marlin species to changing environmental conditions. The study will also include a geographical analysis to identify regional variations in the impact of climatic oscillations. The findings are expected to have significant implications for marine conservation strategies, helping to predict and mitigate climate change impacts on marlin populations and ensuring their sustainability for future generations.” Line 666-677.

---

## [Editor Report · Decision Letter 2]

10 Jul 2024

Can the delayed effects of climatic oscillations have a greater influence on global fisheries compared to their immediate effects?

PONE-D-24-14238R2

Dear Dr. Lee,

We’re pleased to inform you that your manuscript has been judged scientifically suitable for publication and will be formally accepted for publication once it meets all outstanding technical requirements.

Kind regards,

Claudio D'Iglio, Ph.D.

Academic Editor

PLOS ONE

Additional Editor Comments (optional):

Dear Doctor Lee,

The manuscript has been significantly improved according to the reviewers and the editor suggestions.
---

## [Editor Report · Acceptance letter]

20 Jul 2024

PONE-D-24-14238R2 

PLOS ONE

Dear Dr. Lee, 

I'm pleased to inform you that your manuscript has been deemed suitable for publication in PLOS ONE. Congratulations! Your manuscript is now being handed over to our production team.

Kind regards, 

on behalf of

Dr. Claudio D'Iglio 

Academic Editor

PLOS ONE